# What Is It about Art? A Discussion on Art.Intelligence.Machine.

Frederic Fol Leymarie [1,*,†]  and Seymour Simmons III [2,*,†]

1   Department of Computing, Goldsmiths, University of London, London SE14 4NW, UK
2   Department of Fine Arts, Winthrop University, Rock Hill, SC 29733, USA
*   Correspondence: ffl@gold.ac.uk or ffl@creativemachine.org.uk (F.F.L.); seymoursimmons@gmail.com (S.S.III)
†   These authors contributed equally to this work.

**Abstract:** The interrelationship among art, intelligence, and machine has important implications for the visual arts as part of a general education. Here, Frederic Fol Leymarie (FFL), a computer scientist and engineer at Goldsmiths College, and Seymour Simmons III (SS3), an artist and art educator from Winthrop University, South Carolina, discuss these issues and the value of sustained cross-disciplinary conversations in addressing challenges in the 21st century.

**Keywords:** Homo sapiens sapiens; cognitive horizon; embodiment; visual arts; artificial intelligence; perception; drawing; representation; robots; education; mind-body problem; reflection; machine art



## 1. Introduction

### 1.1. Introduction by Seymour Simmons III (SS3)

In the fall of 2011, when I was on sabbatical at Columbia University studying recent research on drawing and cognition, Columbia sponsored a symposium,[1] "Thinking through Drawing" (Kantrowitz et al. 2012). A highlight was Paul, a drawing robot that made portraits. Paul was a collaborative creation of Frederic Fol Leymarie, a computer scientist and engineer, and Patrick Tresset, an artist who at that time was a doctoral student under Frederic's supervision. My communication with Frederic continued and his research influenced my recent book, *The Value of Drawing Instruction in the Visual Arts and Across Curricula: Historical and Philosophical Arguments for Drawing in the Digital Age* (Simmons 2021). He opened my eyes to the potential of digital technology to extend drawing's impact on seeing, thinking, and learning, issues confronted in Frederic's 2021 article,[2] "Art.Machine.Intelligence" (Leymarie Fol 2021). His invitation for feedback provided an opportunity to renew our dialogue which here we open to a wider audience.

### 1.2. Introduction by Frederic Fol Leymarie (FFL)

Seymour and I have been conversing for over a decade on topics relating to learning, expressive drawing, visual perception, the impact of new technologies on our practices and thinking, and the philosophy of artificial intelligence. More recently, we have focused on the importance of the arts for education, learning, scientific discovery, archiving of knowledge, exploration, and the imagination. After our initial encounter, we met again in late May 2013, at the symposium entitled "Drawing in the University Today" (Almeida et al. 2014) held at the University of Porto, where I gave a keynote on drawing robots and perception. We discussed various topics and learned from each other. He inspired me to continue my discussions about art and science with colleagues in the arts and humanities that began during my doctoral studies in Engineering a decade earlier (Leymarie Fol et al. 2001; Hatcher et al. 2005). Perhaps my first realisation regarding the potential for discovery and deeper understanding by engaging with the "other" was my reading in the early 1990s of Rudolf Arnheim's "Art and Visual Perception" (Arnheim 1974).

Here, Seymour (SS3) and I (FFL) summarise discussions about art, perception, intelligence, machines, computing and artificial intelligence (AI) that we have had in re-

cent years, conversations brought to a head by Seymour's beta reading of my essay "Art.Machine.Intelligence" in which I summarise my views.

## 2. About Human Intelligence and the Place of Art

**SS3:** To begin, I have a question about the title of this article and your previous essay: how do you define "intelligence" in a way that connects it to art and machines?

**FFL:** I base my definition of intelligence on the human capacity to act in and react to the world. I consider that various levels of intelligence are present in all species. Humans arguably have a more diverse set of capabilities than other species.

This definition relates closely to the notion of levels of intelligence understood as a function of a system's "cognitive horizon" (Levin and Dennett 2020), which encompasses how to detect, represent as memories, anticipate, decide among and—crucially—attempt to affect experiences.[3] It applies to all cognitive systems, including those of animals, cells, synthetic life forms, and AI, and includes both a spatial dimension—how far away an agent can sense and exert actions, and a temporal dimension—memory and anticipation.

This notion necessitates a body that senses and acts. It corresponds to the main form of intelligence considered in modern robotics and AI, although the current focus of machine learning often neglects the embodied role of the agent. "Cognitive horizon" provides a basis for defining levels of intelligence but does not focus on the qualities that differentiate humans from other species.

I want to focus on the human qualities and abilities that do not seem to be shared by other species, at least not to such a high degree. These include art, architecture, language, mathematics, the making of tools (encompassing machines, computers, and AI), music, writing systems, archiving knowledge, and science. What also distinguishes humans is their capacity to increase their cognitive horizon throughout history. Artistic practice stands at the emergence of *Homo sapiens sapiens* and is likely one of its defining factors.

**SS3:** While these human capacities are significant, it's striking that you, as an engineer and computer scientist, focus on the visual arts as an especially critical domain of intelligence today. Beyond our personal interests in this domain, I think we agree that the power of visual technology and media has made the visual arts a dominant force for good and ill in contemporary culture. In my view, that makes it more essential that everyone—not just artists and designers—receive visual arts education. In this regard, its useful to compare your views to Howard Gardner's (Gardner 1983) theory of multiple intelligences (MI).[4] To begin, Gardner defines intelligence as: the capacity to solve problems or make products of value in one or more cultural settings (Gardner 1983, p. x). This definition differs from intelligence as defined by standard IQ tests, which focus exclusively on linguistic and logical/mathematical abilities. Gardner's more inclusive definition of intelligence emphasizes the arts as domains of intelligence, just as yours does. On the other hand, one of his criteria for a particular capacity to count as a full-fledged "intelligence" is that it can be found in non-human species. This may be distinguished from your concerns, above.

**FFL:** My focus is complementary (if not contradictory) insofar as I focus on capacities not found in other species except to a limited degree, such as the use of tools by some bird species. This is because I am interested in what distinguishes humans from other life forms.

Although elements of such skills emerge to a lesser degree in other species, I focus on what makes humans unique. The creation, design, and practice of the arts is unique to (or at least uniquely developed by) humans. From an evolutionary perspective, it ought to signal a fundamental break from all other species up until *now*—i.e., until the emergence of the *Homo sapiens* capable of producing, enjoying, sharing, and archiving art.

**SS3:** I have a similar opinion about drawing, that it is one of the few uniquely human capacities. Although it is true that many creatures make intentional marks, and that some, like elephants, can be trained to make representational images, only humans have developed the range of skills addressed in my book: drawing from imagination and observation, as well as using drawing for creative problem solving, and as a vehicle for the expression of feelings and emotions. Moreover, drawing is the original graphic symbol

system and the root of later systems including writing, numbers, and musical notation. Thus I find it ironic that formal drawing instruction is now largely neglected as part of childhood education, while so much time and effort are placed on learning how to write.

Another factor related equally to drawing and writing that distinguishes human cognition is the capacity to reflect upon actions, feelings, and thoughts, to evaluate them and to find meaning in them beyond their practical outcomes. Your recent essay (Leymarie Fol 2021, p. 2) directly deals with reflection in relation to art. Please explain what you mean by "reflection" in this context and how it plays out in AI.

**FFL:** In that essay, I define reflection as referring to:

the perception necessary for creating and observing artefacts. This act implies: decisions and comparisons; the planning of actions and movements; the use of tools and materials; the historical legacy that influences an artist's style and exploration of new ideas (acquired mainly during schooling, education or while in apprenticeship in an artist's studio); artists' memories of their own previous creative experiences and past production; the cultural melting pot in which we live and that influences an artist's creativity; the meanings and discourses associated with artefacts; and the interpretations and evaluations of observers.

Reflection is a challenge for AI research. There are no existing AI entities, nor any AI projects that address reflection as an innate skill, although we are still in the infancy of AI research and development. A historically important example of an early AI entity having creative skills is AARON, designed by the late Harold Cohen (McCorduck 1991), although it is not capable of reflection. AARON was conceived as an expert system, in which Cohen would integrate increasingly sophisticated rules and skills. Today, the advanced "AI-art" systems rely on machine learning techniques in which machines acquire knowledge from examples of artistic processes and outcomes. Still, such systems are limited in their capacity to judge the outcome of their own production; they tend instead to produce artefacts with distinct styles that are favored by their human masters.

By identifying singular human traits such as reflection, we can make real advances in AI research. AI research attempts to jointly develop machines able to act in the world with human capabilities, while offering new ways to study and better understand human intelligence itself.[5]

**SS3:** Those last points are crucial to our whole discussion. I am glad you give reflection a separate heading because it is not always highlighted as a distinct and integral facet of art making. This is one contribution of Project Zero (PZ) on which I collaborated, "Arts PROPEL" (Winner and Simmons 1992).[6] PROPEL is an anagram for Production, Perception, Reflection, and Learning. The project studied how these dimensions of learning occur optimally in visual arts, music, and imaginative writing. Further, it sought to identify and advance strategies for assessing each aspect of arts learning in K–12 education.[7] Reflection was assessed in sketchbooks and journals. Of course, there is much more to reflection than just assessment; its relation to perception and production is not always easy to parse. How do they relate in your approach?

**FFL:** For me, reflection intersects largely with the world of perception (by the maker/ artist and the observer/critic), while production focuses on actions necessary to achieve a goal.

**SS3:** That view of reflection makes the most sense in terms of the steps in an art project referred to as "formative" and "summative critiques". The former refers to a pause to discuss the work in progress. The latter happens when the work is finished. Beyond such formal stages, however, the relationships among perception, production, and reflection remain difficult to untangle. For example, John Dewey, in *Art as Experience* (Dewey 1934), highlighted the judgment artists make with each brush stroke about how to proceed. Also, artists differ in their processes. Some spend more time sitting back and reflecting about their artwork than painting, while others barely pause from start to finish. In either case, what is important is the evidence of intelligence at work in artistic endeavors. In fact, that is exactly how Dewey describes such processes:

Because perception of relationships between what is done and what is undergone constitutes the work of intelligence, and because the artist is controlled in the process of his work by his grasp of the connection between what he has already done and what he is to do next, the idea that the artist does not think as intently and penetratingly as a scientific inquirer is absurd. A painter must consciously undergo the effect of his every brush stroke or he will not be aware of what he is doing and where his work is going. Moreover, he has to see the particular connection of doing and undergoing in relation to the whole that he desires to produce. To apprehend such relations is to think, and is one of the most exacting modes of thought. (Dewey 1934, p. 45)

### 3. Body–Mind

**FFL:** In discussing intelligence and following in the footsteps of Spinoza (Damasio 2003), Dewey, A. Damasio (Damasio 1994), and others (Pfeifer and Bongard 2006; Barrett 2011; Howard et al. 2019), I make no a priori separation between mind and body.

Intelligence requires a body and exists only in living beings. Higher levels of intelligence occur in animals, whose bodies give them greater capacities to explore and act on their environment than do plants, fungi or bacteria.

Especially in humans, intelligence is expressed by actions and externalised via art practices such as designing, painting, sculpting, sketching, and writing. Intelligence is due to and constrained by our bodies' capacities, which can be augmented through the use of tools and machines.

Machines can play the role of catalyst between art and intelligence. In this context, I consider contemporary machines both from the point of view of sophisticated tools and as the potential embodiment of AI.

**SS3:** When it comes to drawing, one example of that potential embodiment is the robot, Paul. As illustrated in Figure 1, Paul completed a still-life based on a La Fontaine fable, drawn in the style of Patrick Tresset.[8] Paul combines an electronic eye connected to a computer that is connected to a mechanical arm holding a ball-point pen. As the eye scans the subject, the arm holding the pen swings around and around rapidly and apparently randomly, but in the end, the subject matter emerges vibrantly from the scribbles. What were your aims in this project?

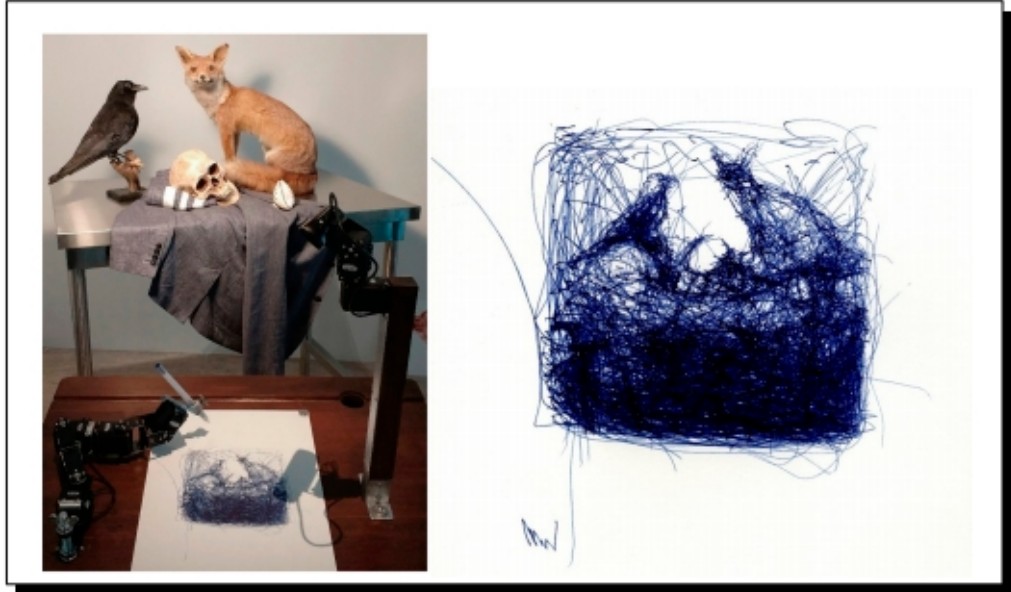

**Figure 1.** Human Study #2, installation with drawing robots and still life. Image courtesy of Patrick Tresset.

**FFL:** Tresset and I wanted to develop a method for exploring artistic skill that could be used to test various hypotheses and ideas. The method consists of several steps. First, in concert with the artist/expert, and through a survey of existing literature, we designed a system intended to imitate the way artists work, for instance a description of steps to achieve a portrait in a given style (Figure 2). Second, we identified the processes for which we required computer programs able to take some input and produce an artefact; e.g., starting from an image of a face that is subsequently transformed, via a sequence of (digitised) drawing actions, into a final portrait. Once the results are of an acceptable quality and resemble the original work of the artist, we proceed with the embodiment into a robotic platform. Once this is achieved, various experiments regarding the perception of machine art (Chamberlain et al. 2017, 2022), or performances thereof, can be conducted. This method proved successful with Tresset, the AIkon project, and the resulting first few generations of Paul the Robot (Tresset and Leymarie 2005; Tresset and Leymarie 2013).[9] We applied this method to graffiti art, calligraphy, and the AutoGraff project on which we collaborated with Daniel Berio (Berio et al. 2016; Berio 2021).[10] The approximate reproduction of the artist's working process is considered a dynamic system. Our goal is to simulate the steps, actions and analyses humans go through when producing an artefact, a simulation that attains increasing levels of sophistication as we progress in our understanding of how artists function.

**SS3:** In your 2021 article, you explain the evolution of the AIkon project. What impressed me most was the concluding paragraph in which you discuss your "first attempts to introduce direct visual feedback. Like the human artist, Paul the Robot could observe its strokes and gestures via its camera-eye, which, till then, had been used only to take a snapshot of a face of someone nearby. Its artificial brain could evaluate in real time the quality of what was drawn in order to decide on what it was to concentrate its attention next while drawing" (Tresset and Leymarie 2013).

In my book (Simmons 2021, p. 194), I compare Paul's process to gesture drawing, a well-known exercise often used for sketching animals and humans (Figure 2); it can also be applied, as here, to a complex composition (Figure 1). When a human draws, the process exemplifies embodied cognition in multiple ways. Allowing the eye to scan freely with the hand following it as quickly as possible makes sense if the subject is a living person or animal, but in Kimon Nicolaïdes' *The Natural Way to Draw* (Nicolaïdes 1941), the process is illustrated by a ribbon tied into a bow, comparing a contour drawing of the subject to a gestural sketch (Figure 3). Both drawings could be considered embodied because, according to Nicolaïdes, in contour drawing the students imagine the pencil touching the edge as they draw, moving slowly and carefully, while in gesture drawing, they try to feel the movement of the object as a whole, which means drawing quickly and energetically.

That quality of touch distinguishes for Nicolaïdes contour from outline. In his view, an outline is flat and consists of the edge between two shapes, an object and background, or between a positive shape and a negative space. By contrast, contour implies volume. The experience of drawing an outline versus drawing a contour is significant, like the difference between touching a painting or photograph of a figure on a flat wall versus touching the edge of a sculpture or actual person.

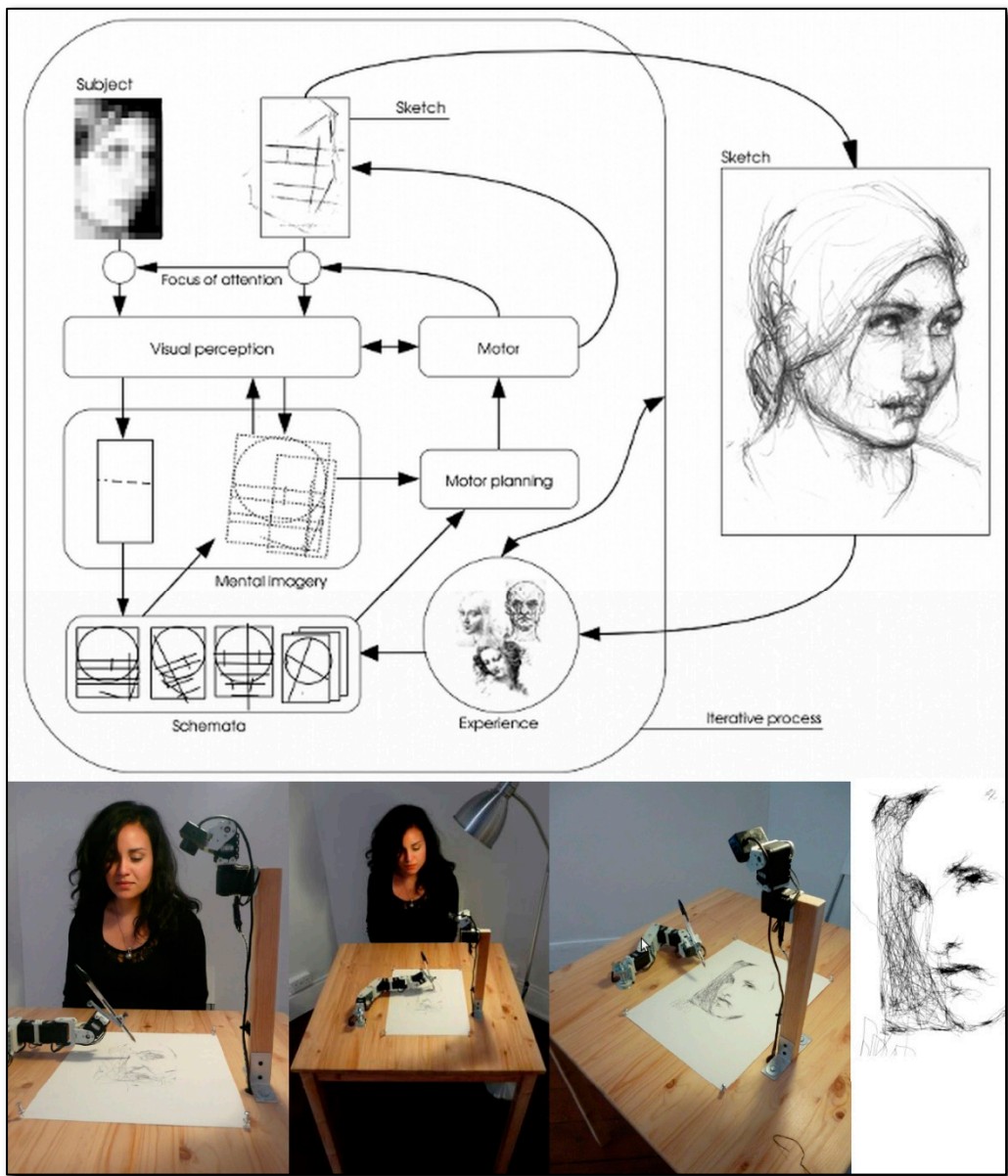

**Figure 2. Top**: Example of a high-level system view of the processes and their interaction necessary to study, simulate, and reproduce the activity of an artist—here in the context of a portrait drawing for the AIkon project (circa 2010). **Bottom**: An example of the embodied system—aka Paul the Robot—performing a live portrait.

Another aspect of embodiment in gestural drawing, especially when the subject is a human figure in an expressive pose, addresses the feeling or emotion embodied in the pose. Figure 4 illustrates that process. The figure is an amateur musician playing a street organ, but the musician is performing in France, where the instrument is called an *Orgue de Barbarie*. I tried to let my hand follow the movement of my eye as closely as possible, while letting my "pencil swing around the paper almost at will, being impelled by the sense of action" (Nicolaïdes 1941, p. 14). Moreover, while my eyes followed the action of the musician, my marks were also impelled by the rhythm of the music as well as by what I felt kinesthetically and emotionally to be the spirit, the energy, and the overall personality of the performer.

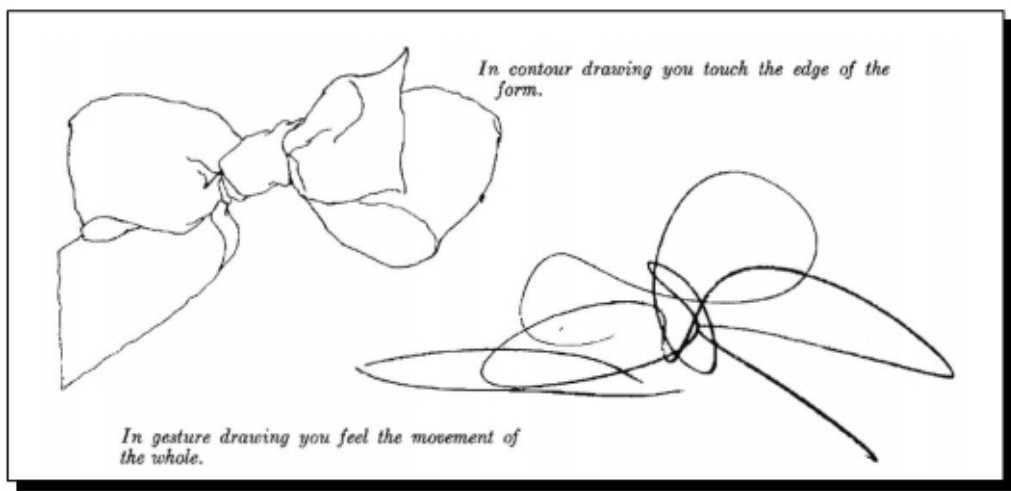

**Figure 3.** From "The Natural Way to Draw", by Kimon Nicolaïdes (1941).

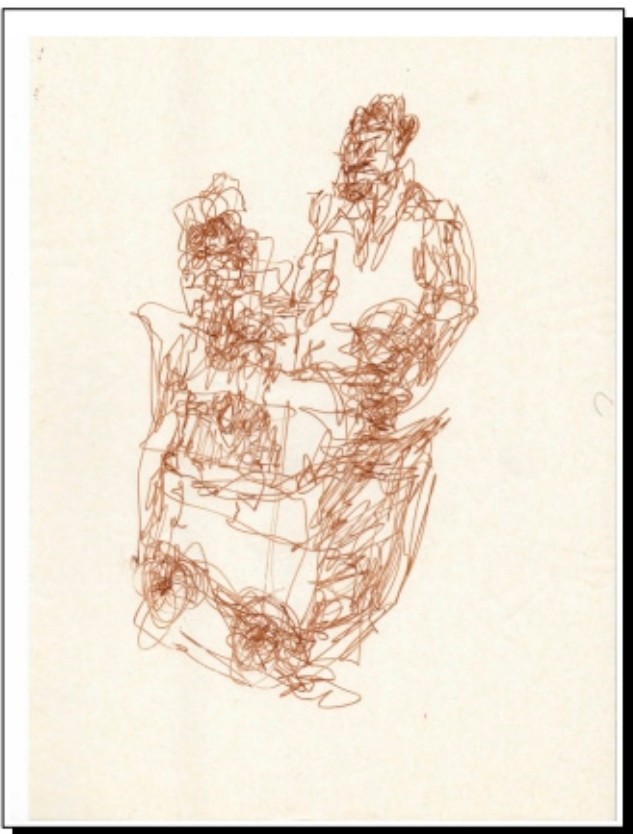

**Figure 4.** *Orgue de Barbarie* by Seymour Simmons III; brown pen on acetate (2018).

In art, as in life, the ability to interpret postures and facial expressions in terms of feelings seems to come intuitively. But people differ in their ability to "read" one another, and the skill can be developed. In learning to draw from the model, we acquire that awareness because the first attempts are usually inaccurate and convey a different feeling or expression. An analysis of the drawing allows us to identify and correct the problem. In that way, we learn to draw more accurately and to see more sensitively, potentially cultivating more empathy. Applied to drawing oneself in a mirror or just through sensation, quick sketches like this can contribute to kinesthetic and emotional self-awareness, and ultimately self-knowledge.

Damasio gives us a clue to how this happens in *Looking for Spinoza*, where he attempts to resolve the famous philosophical challenge, the "mind–body problem, a problem central to the understanding of who we are." In the process of connecting mind to body, he also connects thought to action through emotion and feeling, which are not the same thing:

> Emotion and related reactions are aligned with the body, feelings with the mind. The investigation of how thoughts trigger emotions and of how bodily emotions become the kinds of thoughts we call feelings provides a privileged view into mind and body, the overtly disparate manifestation of a single and seamlessly interwoven human organism. (Damasio 2003, p. 7)

While Damasio looks at mind–body questions from a 21st century neurological point of view, he also looks back to the 17th century philosopher Baruch Spinoza for an explanatory framework. Since Spinoza supposedly kept a sketchbook, he may have developed his ideas about mind–body relationships through drawing (Berger 2011). Two 20th century philosophers have also tackled this subject: John Dewey, in *Experience and Nature* (Dewey 1925) and Gilbert Ryle, in *The Concept of Mind* (Ryle 1949). The following is a summary of Dewey's theory.

Dewey felt that the conception of mind should be a verb:

> Dewey rejected both traditional accounts of mind-as-substance (or container) and more contemporary schemes reducing mind to brain states. Rather, mind is activity, a range of dynamic processes of interaction between organism and world. Consider the range connoted by mind: as memory (I am reminded of X); attention (I keep her in mind, I mind my manners); purpose (I have an aim in mind); care or solicitude (I mind the child); paying heed (I mind the traffic stop). "Mind", then, ranges over many activities: intellectual, affectional, volitional, or purposeful. [Mind] is "primarily a verb . . . [that] denotes every mode and variety of interest in, and concern for, things: practical, intellectual, and emotional. It never denotes anything self-contained, isolated from the world of persons and things, but is always used with respect to situations, events, objects, persons and groups". (Hildebrand 2018)[11]

Ryle critiques the Cartesian mind-body dichotomy in which the former is essentially spiritual, and the latter is mechanical, e.g., the "ghost in the machine," and with no clear explanation of how they interact. This he explained in Chapter II, "Knowing How and Knowing That", where he tried to "show that when we describe people as exercising qualities of mind, we are not referring to occult episodes of which their overt acts and utterances are effects; we are referring to those overt acts and utterances." (Ryle 1949, p. 25).

Ryle explains the problem in terms of a "category error" in which the mind and body are mistakenly lumped into the same logical category but actually belong in different ones, for example, when one visits Oxford and sees its buildings and structures, then asks "Where is the university?" thinking it must be another one of those distinct entities. Just as "university" is the overarching entity within which individual entities are embedded, likewise, "mind" encompasses actions, utterances, etc. Mind could be embodied intelligently in action, but not if we react without thinking. Even so, according to Ryle, thinking could be part of the act, not preceding it.

This aligns well with Dewey's conception of "mind" as a verb. Activities can be done "mindfully" or "mindlessly", i.e., thoughtfully or mechanically. Similar issues apply to AI.

**FFL:** I would like to extend our discussion by referring to Dewey's four claims regarding the necessity of embodiment (Vaesen 2014):

1. "Cognition should be studied in the light of the evolutionary processes that shaped it."
2. "Cognition is embodied and extends into the world."
3. "Cognition is active and practical, rather than passive and theoretical."
4. "Cognition is situated."

According to Point 1, cognition links to evolutionary principles, to how adaptation permits the transfer of useful features for a given species and lineage. Point 2 aligns

with the embodiment hypothesis by implying the role of the environment in which the body-mind acts. Point 3 emphasizes the role of actions or reactions (e.g., in movements, search, manipulation). And Point 4 asserts that the organisms must be studied together with their environments.

Dewey's philosophical stance provides a basis for a discussion on the relationship among art, intelligence, and machine. Art is an activity performed by the body and constrained by our manual capabilities.[12] Art is also directly influenced by environmental conditions: for example, in the types of surfaces and materials used by a graphic artist, from cave painters to modern graffiti artists. We move and act in the world with our articulated bodies, manipulating objects (brushes, carving tools) in exploratory modes.

Figure 3 relates contour drawing to touch. As John M. Kennedy in *Drawing and the Blind: Pictures to Touch* (Kennedy 1993) observes, a blind person can accurately draw objects they have experienced mainly by touch (Figure 5). The outcome of such drawings is often similar to outline drawings by the sighted. Kennedy proposes that what is drawn directly represents what is felt by touch: the main features of an object, its ridges, crevasses, mounts, and such, memorised by touch. This situation raises questions about how we think about an object's fundamental structure. Kennedy argues for a unified theory of human perception that combines sensing by sight with sensing by touch; the unification apparent in how shape is represented.

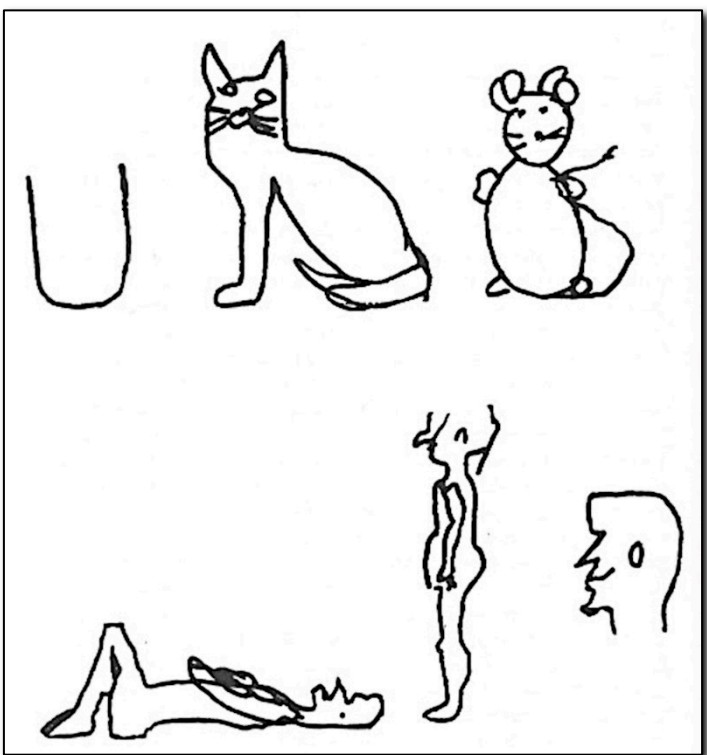

**Figure 5.** Drawings (a glass, a cat, a mouse, a man lying down and standing up, and a head) by Tracy as an adult, and blind since age two. (Kennedy 1993, Fig. 4.16, reproduced by permission of the author).

Figure 4 exemplifies the power of the hand in action. What we see is not a drawing made of lines or even curves, but an assembly of strokes (Chabrier 2021), the traces left by gestures. "Gesture" relates to the body engaged in intentional action in which traces are linked to eye movements. Imprecisions are important because they convey volume and can evoke movement—the musician performing on the organ—but also provide cues to the process used to realise the drawing itself. This illustrates how the stroke is a much richer entity than the mathematical objects defined as (smooth) curves.

You asked about AI. Cognition in current AI research is largely considered from an abstract computational point of view with little consideration of the impact of a body and its potential—e.g., the "brain in a vat" or "disembodied cognition". This community is engaged in an on-going debate regarding the form of computational modelling that should be used, such as connectionist versus symbolic representations. *Connectionist* refers to computing architectures that use data to produce outcomes, for example by training artificial neural networks to adapt to (or learn from) examples in order to produce reasonable responses to new instances. *Symbolic* refers to intermediate representations upon which the computational architecture revolves; a traditional example would be an "expert system," which consists of rules intended to capture sufficient knowledge from an expert to enable it to answer questions or address problems. Some argue for closer integration of these two ways of modeling computational architectures in order to create hybrid solutions. This overlooks, however, the relationships between cognition and the body and its potential for action.

**SS3:** Kennedy referred to drawings by the blind as outlines, not contours, and I agree. Sighted people see outlines based on shape and spatial relationships, which makes observational drawing easier. By contrast, the blind must sense volumes as a whole, then create an outline from their holistic understanding, which is more difficult. So even though the guiding sensation comes from touching an object—or if they are drawing themselves, from proprioception—it is still not the same as the tactile experience, the "contour drawing" Nicolaïdes discusses. In fact, both contour drawing for the sighted and outline drawing for the blind relate to your question about "cognition, the body, and its action potential."

For one thing, your discussion of "exploratory modes" evokes Dewey's concept of intelligence. Greater perspective on this issue comes from contrasting Dewey's epistemology with the classical schools: Rationalism and Empiricism. Compared to Rationalists like Plato and Descartes who believed that knowledge was derived from principles preexisting in the mind, and Empiricists, like John Locke, who assumed knowledge was gained directly from sensory experience imprinted on the mind, Dewey and his predecessors C. S. Peirce and William James defined their philosophy as Pragmatism, a theory of knowledge in which we come to know something by the operations we perform upon it. This is what Dewey refers to as the interaction between doing something and experiencing the consequence of that effort, making action and cognition interdependent. This applies to the role of drawing in learning to observe in which we compare what we draw to what we see, in the process of which we notice differences and observe more accurately. For me, epistemology helps explain how teaching drawing in different modes involves different modalities of thinking, which in turn connect art with disciplines like science and mathematics.

## 4. The Fundamentals of Art

**FFL:** I want to focus on the foundations of art: that which we all experience as youth, and which is necessary to develop our intellect as well as creative practices: skills, materials, representations (as patterns on flat canvas, sculpted, in architecture, in landscaping, music).

Those fundamentals are different from other elements often assumed to be relevant when discussing art: such as economic value, tastes, cultural influences, social and political statements made through art.

I do not consider such aspects as fundamental to art in relation to the emergence of *Homo sapiens sapiens* as distinct from other species. I consider them separate from what the essence of art is; they arise later in human history. These fundamentals of arts should be taught from pre-school to university. The rest is a matter of taste, secondary in importance and not crucial to our study of art, intelligence, and machines.

**SS3:** I agree with you about fundamentals, although I believe this topic should be described as analysis, because such fundamentals are applicable to making art and to analyzing works of art to understand how they represent, express, and convey meaning. But "fundamentals" is a contentious topic in art education, especially at the post-secondary level, where the overarching question is: who decides what counts as fundamental.

Traditionally, the preparation of artists involved mastering long established skills and techniques, especially through copying. This was largely abandoned in the 20th century under Modernism. Instead, individuality, creativity, and self-expression has been encouraged. While such aims are laudable, the result is that many skills and bodies of knowledge once common to artists, for example linear perspective, are now almost lost, which leaves many students, even in art schools, assuming they cannot draw.

I argue (Simmons 2021) that fundamentals are essential but should not be limited to those traditionally taught. For example, rather than learning only linear perspective, a Western European tradition, 21st century students should learn a range of "projection systems" used in various cultures that are applicable to drawing problems outside the realm of art, for example in engineering, architecture, and design.

In art education, perspective and "fundamentals" are usually covered under "the elements of art" and "principles of design".[13] I addressed this in the chapter "Languages of Drawing," since elements and principles function as grammar and syntax in ordinary language.[14] Philosophically, they fall under semiotics, since drawing and writing are both graphic symbol systems. You address such concerns, along with skills and style, in your research on graffiti and pen and ink drawing.

**FFL:** Yes, I highlight experience, knowledge and techniques accumulated over centuries. In the works with Patrick Tresset (projects AIkon and AIkon 2),[15] the focus was placed on techniques including: the use of shaded areas to convey shape, rapid gestures with varying degrees of ink to fill-in such areas, the use of certain delineating contours (of the face), and the use of white (or empty) areas to emphasise form and give depth (Figure 6). In the ongoing study with Daniel Berio and our collaborators—in the AutoGraff project—our focus shifted to the importance of the kinematics implied by rapid and skilled movements when performing graffiti art.[16] There, we study how speed and acceleration of limbs and hands perform traces, which when seen by observers, exemplify aesthetic qualities (Figure 7). The underlying hypothesis is that the visual aesthetic (our appreciation of a graphic outcome) is in part dependent on movement recoverable visually by the observer of a graphical artefact *a posteriori*: even the naive observer can imagine how traces were produced and reflect upon the naturalness or the skills needed to produce them (Freedberg and Gallese 2007; Leder et al. 2012; Berio 2021).

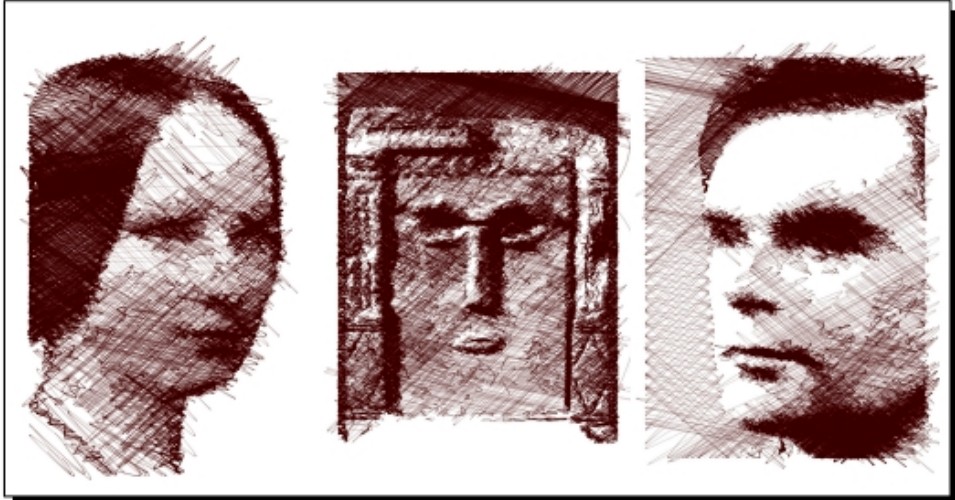

**Figure 6.** Examples of AIkon 's rapid sketching of shaded areas. **Left**: Ada Lovelace, early pioneer of the concepts of software and programming, 19th century; **Center**: Eye idol from the Temple of the Winged Lions, Petra, Jordan, 1st century (Markoe 2003); **Right**: mathematician and computing pioneer Alan Turing, 20th century.

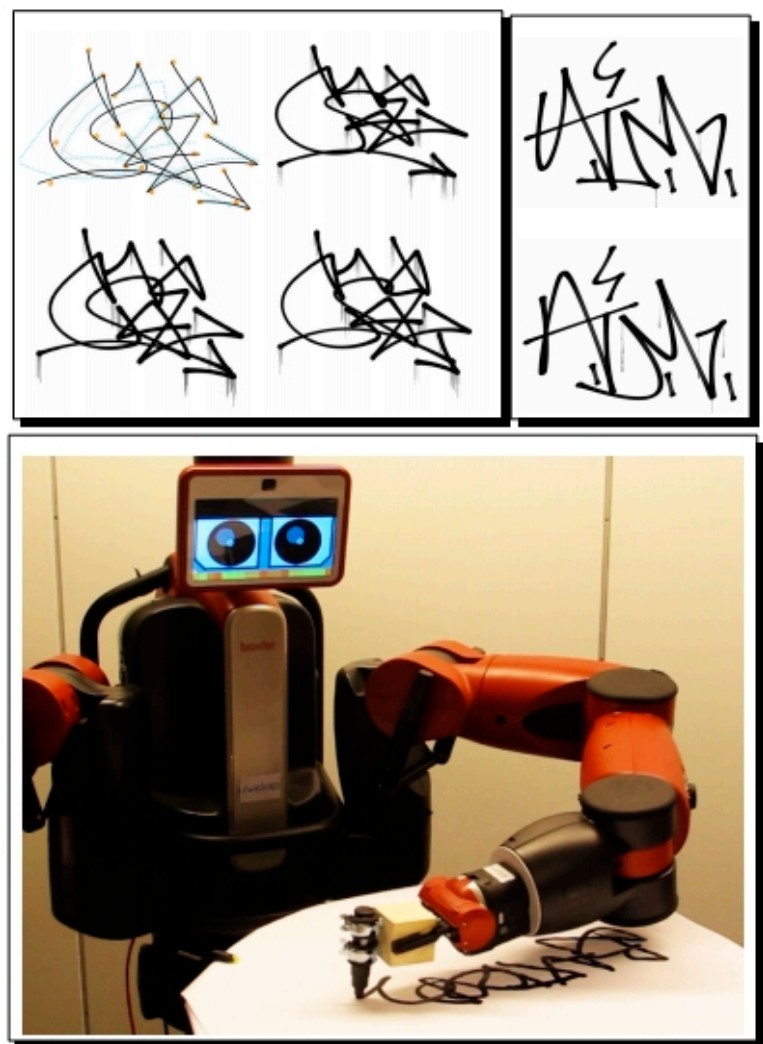

**Figure 7.** Examples from the AutoGraff project. **Top-Left**: graffiti tags with same targets or aiming loci (yellow dots, top-left) and 3 variants obtained by changing the speed profiles of each gesture (one per pair of successive targets). **Top-right**: Two versions of tags (with same targets, but different speed profiles) for "A.I.M.". **Bottom**: The Baxter "compliant" robot capable of writing its own name in the style of graffiti tags (Berio et al. 2016).

Having a reference work on the topic would be useful. Rudolph Arnheim, a student at the early 20th century Gestalt school, described a subset of such know-how in the arts (Arnheim 1974), with a focus of physics-based concepts applied to perception.

**SS3:** You are right about the viewer imagining how the marks were made in a drawing. This is one reason why people enjoy drawing. As well as appreciating the subject matter, they enjoy the sense of how the image was produced. About your other point, Arnheim and the Gestalt school would be great resources. Their analysis of works of art according to elements and principles helps explain how art, including abstract and non-objective art, conveys meaning. These formal factors also apply to making evaluative judgements.

Speaking of elements and principles, we should not forget the importance of medium and style, both significant in terms of semiotics. To cite Marshall McLuhan, "The medium is the message" (McLuhan 1964), and that applies to graffiti artists as well as fine artists, designers, and illustrators. Thus, art students as well as professionals spend quite significant time experimenting with media before settling on a particular style. Some continue experimenting for its own sake, without settling on a particular style. Figures 8–10 were executed when I was working as an illustrator and looking for varied projects. They all involve animals but exemplify different conceptions of drawing: "drawing as design"

(Figure 8), "drawing as seeing" (Figure 9), and "drawing as expression" (Figure 10). Also, all were experiments in media and fit under "drawing as experience and experiment" and employed various mark-making systems that could be categorized as "languages of drawing". That does not necessarily imply that artists are always consciously using such factors as they work. By contrast, some suggest that cave paintings were based on dreams (Lewis-Williams 2002), Figure 11.

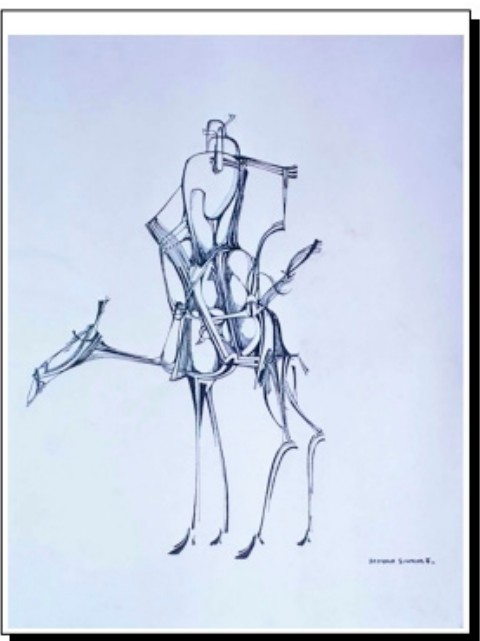

**Figure 8.** "Don Quixote I" by Seymour Simmons III; an example of "drawing as design".

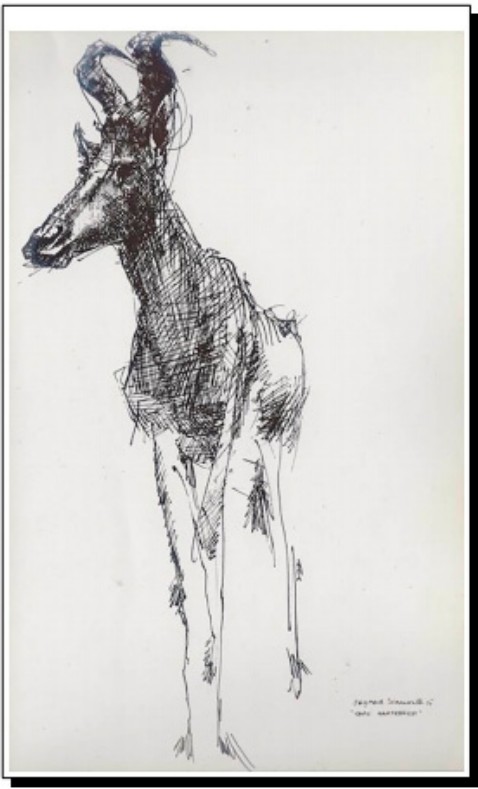

**Figure 9.** "Hartebeest" by Seymour Simmons III; an example of "drawing as seeing" or observing.

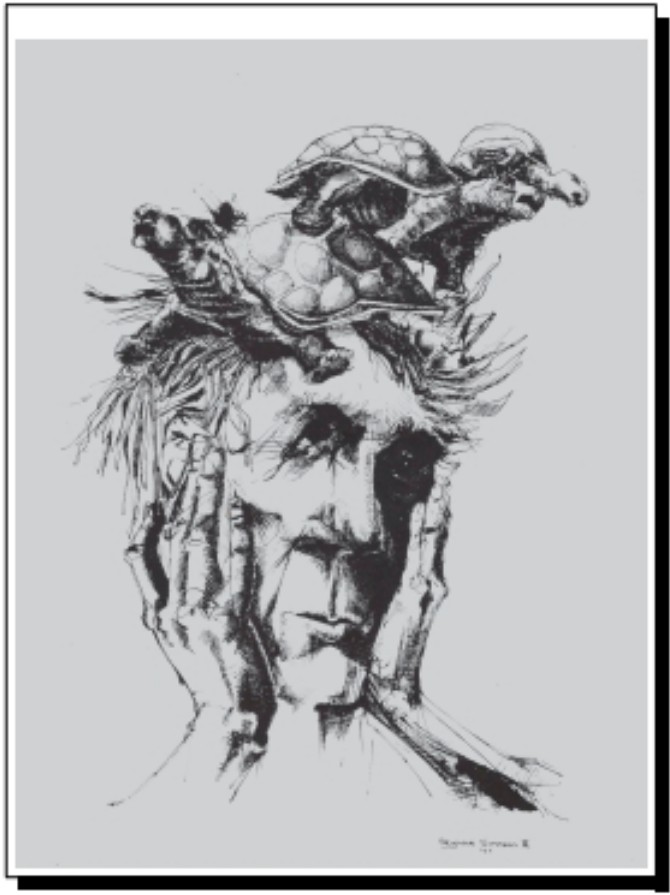

**Figure 10.** "Turtles" by Seymour Simmons III; an example of "drawing as expression".

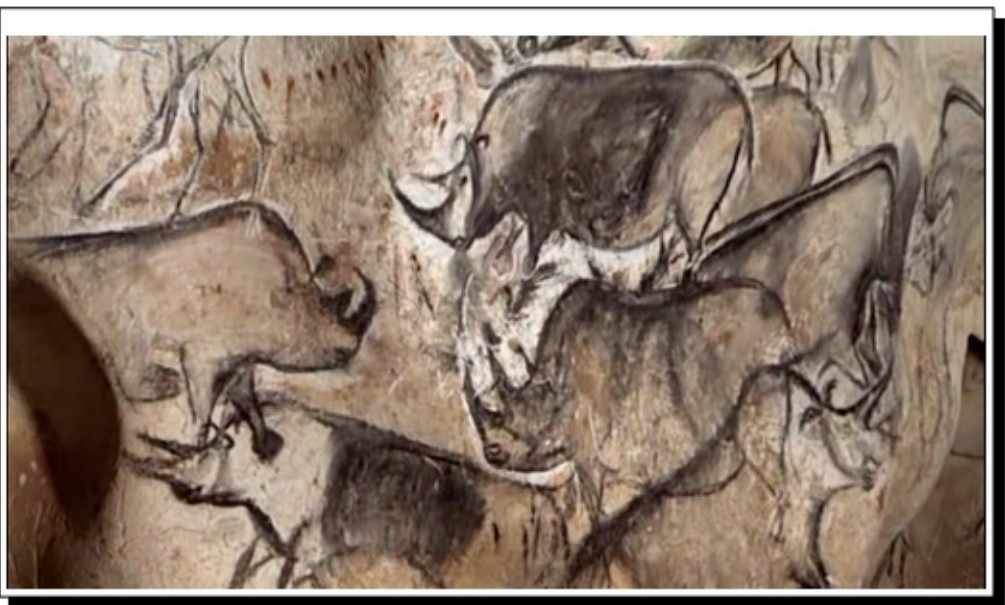

**Figure 11.** Rhinoceros drawings from Chauvet Cave (30,000 BCE).

## 5. On Perception and Creativity

**FFL:** This last point is crucial to our discussion. We must "hallucinate" the world, make sense of it. We are guided by our limited five senses to interpret the world, to give it useful meanings. We have evolved to be hyper-creative, such that we constantly invent new patterns, new ideas, new tools. Diversity of illusions find their place in our

perceptual spectrum.[17] Even when asleep, we dream, inventing a continuously changing virtual theater.

We continuously create meaning for our sensations. This is described in psychology as operating through user interfaces integrated into our nervous and haptic systems (Hoffman et al. 2015; Koenderink 2019). Such interfaces enable us to understand the world in- and outside our bodies. For example, the nervous system invents the meaning of colors as it tries to understand what we have visually sensed. Another species exposed to the same electromagnetic signals will have a different interpretation based on its own color palette (created over evolutionary scales). We associate shapes and functions with perceived objects. For example, various physical objects can be used as a table. These form part of our interface with the world (Leymarie Fol 2011), as distinguished from the object's own physical character.

Artistic creativity represents an evolutionary step in exploiting and deploying the natural need to invent meaning in the world. Without this faculty, we may have been stuck at the level of our primate cousins. Humans pushed their creativity further. And this made all the difference.

**SS3:** To me, this links to the representational dimension of thought, although often a thought is reduced to an easily identifiable schema. In art, picturing a thought constitutes a much richer, more creative, and more individual process.

**FFL:** I see art as a necessary step leading to the invention of more recent methods of creating and archiving knowledge. The emergence of visual art forms can be traced back to the beginnings of *Homo sapiens sapiens* (Bednarik 2003; McDermott 2021). Through the ages, initial visual forms evolve, leading to the introduction of symbols in the form of logograms, which fewer than ten thousand years ago, initiated the practice of writing,[18] and later literature and the archiving of knowledge and scientific discoveries. Mathematics itself required first the invention of symbolic and graphical representations and systems. These systems of communication appear as extensions of the original art forms that preceded them. Thus, from a historical perspective, without earlier forms of art there would be no literature, mathematics, or science, and no accumulation of knowledge.

But it is not so much the historical precedence of art that is fundamental, but rather its manifestation of our thirst for creativity, invention, imagination, dreaming the world and making sense of it. AI will evolve in a major way when it becomes creative and can be aware of its own creations—such that an AI can appreciate, evaluate, and reflect upon it. A non-creative and non-aware AI in this sense remains akin to a zombie, a hollow machine.

**SS3:** These last points are intertwined, and part of the powerful underlying message. For you, the question then turns on technology. For me, the question goes back to how visual arts education can play a part. In this regard, I recall drawing's role in creativity is expressed with the familiar phrase: "back to the drawing board!" Beyond that, following Csikszentmihalyi, we know that creativity involves not only solving problems but finding problems worth solving (Csikszentmihalyi 1990).[19] This returns to my earlier remarks about drawing for ideation and manufacturing in fields like architecture, design, and engineering.

Architecture and design are obvious examples of intelligence in the visual arts. But our interests here are more inclusive and holistic. For example, Antonio Damasio emphasized the relation of emotions to cognition. Likewise, Howard Gardner identified interpersonal and intrapersonal intelligences that relate to understanding other people's feelings and our own. Equally important is aesthetics. Emotions and aesthetics raise questions we cannot fully address here, but I hope there soon will be a way to connect those elements with AI and machines in an evolutionary framework.

## 6. On Art, Intelligence & Machine

**FFL:** One of my basic starting points is that: art offers a window on human intelligence. In addition, art evolves (in its practice, supports, and results) in a symbiotic manner with advances in machines.

**SS3:** This is a critical point because it connects what you write about the arts and what you write about intelligence. You connect these common features of the arts to common features of intelligence.

**FFL:** I tried to summarise this in the title of the essay (Leymarie Fol 2021). I see machines as the most recent evolutionary stage of our tools linking the pair: art and intelligence. Humans developed primitive tools in order to leave traces and archive early forms of knowledge (some of which we now consider as the birth of art). In turn, we refined these tools over generations to expand our intellectual reach. The original traces of artistic practices date back to the first stone engravings (Bednarik 2003). Throughout history, we have developed a capacity to impact and transform materials through the use and evolution of our tools, partly through our artistic activities. This development follows hand in hand that of our intellectual capacities.

**SS3:** Your reference to hand in hand reminds me of *The Hand*, a book by Frank Wilson (Wilson 1999). There, he argues that the brain evolved as a direct consequence of the development and functionality of the hand, so tool-use in art and practical matters was integral to human cognitive development.

**FFL:** Many species have hands. How they use them is to me the significant issue. In the case of humans, their use for the design and creation of tools seems key. For example (Heldstab et al. 2016): "our results support the idea of an evolutionary feedback loop between manipulation complexity and cognition in the human lineage, which may have been enhanced by increasingly terrestrial habits".[20] The development of an artistic practice participates in this co-evolution and underlines it as a key factor that distinguishes humans from other primates.

## 7. Conclusions

**SS3:** One of my goals in this dialogue has been to come to a greater understanding of the potential of machines as complements to traditional artmaking (especially drawing) rather than competing with it. At the same time, my recent book (Simmons 2021) responds to the decline of drawing instruction in schools of art and design, as well as in preK-12 art education, due largely to the power of digital media found in imaging tools used in art and Computer-Aided Design packages used in architecture.[21] I am concerned about the loss of hand-drawing skills in an increasingly digital world, especially for younger learners. However, thanks to conversations like this, I am starting to recognize positive implications of digital media for education. Specifically, the proliferation of imaging technologies, and child-friendly versions, reflects how important visual thinking is today. These facts affirm the alignment of art, machines, and intellect. They also support arguments for giving art a central role in education. Such initiatives are important for drawing since it crosses curricular divides and can contribute to creativity in science as in art. Thus, students at all levels should learn to draw just as they now learn to write: as an art form in and of itself and as a skill that can be applied across curricula. Further, intelligence developed initially through experience with traditional art media such as drawing might transfer to the development of more human-like (and more humane) capacities in AI–infused machines.

This idea was suggested by Glenn W. Smith, co-editor of the ARTS Special Issue: *The Machine as Artist* (Leymarie Fol et al. 2020). In my article for that issue, "Drawing in the Digital Age" (Simmons 2019), I argued that learning to draw "could help us to become more fully human". In response, Glenn referred to the introduction for that issue (Smith and Leymarie 2017), and asked me to consider that "training computers and robots to see and draw sensitively must also take us . . . closer to a 'friendly AI' or to a friendlier technology in general."[22]

I have much to learn before I can make an impactful contribution in this regard. Nonetheless, our conversations and the projects discussed here suggest some possible directions for future research. One is AIkon, which embodies a more responsive and reflective capacity for AI. Another involves taking what we learn from AI and applying it back to human beings and their potential intellectual development.[23]

Both return us to the question of embodied intelligence, in which thinking is not only prior to the act or after the fact, but part of the act itself as discussed by Dewey and Ryle. Likewise, following Damasio, thinking is intertwined with feeling and emotions. Unfortunately, these interdependencies are often neglected in academic education, but they have always been part of education in the arts, which brings us back to the question in our title: What is it about art?

**FFL:** One of my goals has been to convince the scientific community studying forms of intelligence and the high-tech community focusing on AI that art is fundamental to their practice and to the education of the next generations, rather than convince artists that progress in science is relevant to them.[24]

This is a challenge for our time, our visually overloaded culture, filled with information and misinformation: how shall we judge what is true or false, what makes sense? It is crucial that we learn how visual information works with formal means. Modern tools are too powerful to be operated out of ignorance, like monkeys on typewriters writing Shakespeare, but having no idea of how and why they wrote what they did.

Artistic activities provide a high potential to explore creativity, to nurture it. In order to produce new interpretations, ideas, symbols, and knowledge, intelligent systems will need to be creative. Humans are hypercreative in comparison to all other known living species, and we can reasonably identify this trait as the breakthrough in the evolutionary tree that allowed us to become *sapiens sapiens*. If we are to produce breakthroughs in our understanding of human intelligence, and its sources, and further develop AI systems that collaborate in extending our cognitive horizons, we will need to integrate artistic mindfulness in our AI agenda: I propose to refer to such a program as A.I.M. (Art.Intelligence.Machine.; Figure 12).

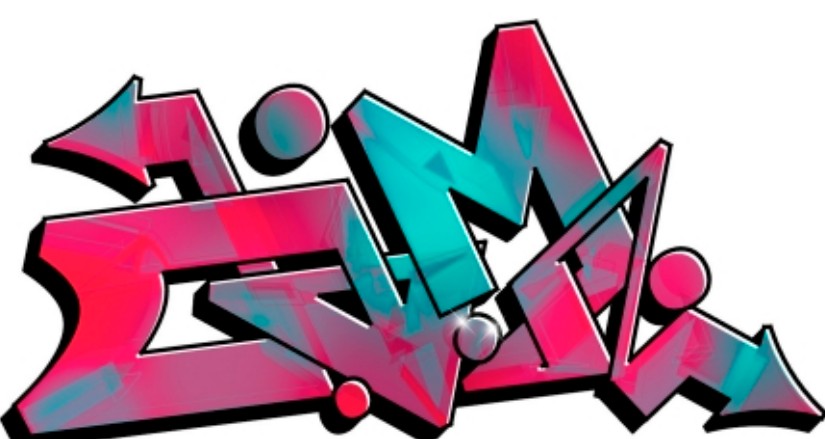

**Figure 12.** A.I.M.: Art.Intelligence.Machine., by Daniel Berio with the AutoGraff system (Berio 2021), a graffiti tag at the intersection of the arrow and machine styles in the words of Dado, an artist and theoretician of graffiti art (Ferri 2016).

Our art, tools and machines have co-evolved with our collective intelligence. Our history is one of continuously extending our cognitive horizons. Art is a distinctive original trait of human intelligence. You ask about what the future may offer us. It seems to me that the exploration of creative forms of AI in the digital as well as through robotics is a promising avenue to explore. I agree that the general educational curriculum should value artistic activities at all levels. How else could we begin to fully understand the origins, development, and future potential evolution of *Homo sapiens sapiens*?

**Author Contributions:** These authors contributed equally to this work. All authors have read and agreed to the published version of the manuscript.

**Funding:** This research received no external funding.

**Institutional Review Board Statement:** Not applicable.

**Informed Consent Statement:** Not applicable.

**Data Availability Statement:** Not applicable.

**Conflicts of Interest:** The authors declare no conflict of interest.

## Abbreviations

The following abbreviations are used in this manuscript:

| | |
|---|---|
| AI | Artificial Intelligence |
| A.I.M. | Art.Intelligence.Machine. an AI research programme for the 21st century |
| AIkon | The Artistic/Automatic Ikonograph project (sites.google.com/site/aikonproject/, accessed on 8 March 2022) |
| K–12 | Years from kindergarten to publicly supported primary & secondary education in the U.S.A. |
| PZ | Project Zero at Harvard (www.pz.harvard.edu, accessed on 28 January 2022) |
| SS3 | Seymour Simmons III |
| FFL | Frederic Fol Leymarie |

## Notes

1  drawingandcognition.wordpress.com/publications/symposia-proceedings/2011-conference-proceedings-publication/ (accessed on 15 March 2022).

2  https://lnkd.in/dimPcmc4 (accessed on 8 March 2022).

3  https://aeon.co/essays/how-to-understand-cells-tissues-and-organisms-as-agents-with-agendas (accessed on 20 January 2022).

4  https://en.wikipedia.org/wiki/Theory_of_multiple_intelligences (accessed on 11 March 2022).

5  " … I believe the attempt to make a thinking machine will help us greatly in finding out how we think ourselves."—Last words by Alan Turing from a BBC radio broadcast, on 15 May 1951, entitled "Can Digital Computers Think?", part of a series of broadcasts by five British computing pioneers (Jones 2004); a re-recording from the original script is available here: https://youtu.be/cMxbSsRntv4 (accessed on 22 January 2022).

6  http://www.pz.harvard.edu/sites/default/files/Arts%20Propel%20-%20A%20Handbook%20for%20Visual%20Arts_0.pdf (accessed on 2 February 2022).

7  K–12: Years from kindergarten (pre-school) of publicly supported primary and secondary education in the USA.

8  patricktresset.com (accessed on 28 January 2022).

9  https://sites.google.com/site/aikonproject/ (accessed on 28 January 2022).

10  https://www.doc.gold.ac.uk/autograff/ (accessed on 28 January 2022).

11  https://plato.stanford.edu/entries/dewey/#Mind (accessed on 12 February 2022).

12  Art making is constrained by "end effectors" used to manipulate tools; most commonly the hand, but also the mouth or foot. In more recent times, humans have started to use machines as body extensions which can even be used remotely to manipulate such tools.

13  Elements typically include line, shape, form or volume, space, value (light/dark), texture, and color. Principles include such things as balance, contrast, emphasis, movement, pattern, repetition or rhythm, proportion, and unity/diversity.

14  "Languages of Drawing: Semiotics and the Search for Fundamentals" (Simmons 2021, Ch. 8).

15  https://sites.google.com/site/aikonproject/ (accessed on 2 February 2022).

16  https://www.doc.gold.ac.uk/autograff (accessed 2 February 2022).

17  https://www.illusionsindex.org (accessed on 2 February 2022).

18  https://www.britannica.com/topic/writing/Alphabetic-systems (accessed on 2 February 2022).

19  www.sciencemag.org/careers/2009/02/perspective-problem-finding-and-multidisciplinary-mind (accessed on 11 April 2022).

20  www.nature.com/articles/srep24528 (accessed on 4 February 2022).

21  PreK–12: early childhood, usually 3-year-olds, through 12th grade students in the USA.

22  www.mdpi.com/2076-0752/6/2/5 (accessed on 2 May 2022).

23  note 6.

24  As in my experience: artists are very often a priori keen in engaging with the latest developments in science and technology; they seek novelty, are naturally open to new ideas, techniques, representations.

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
