# Peer review of "What Is It about Art? A Discussion on Art.Intelligence.Machine."

_arts, 2022_

Round 1

Reviewer 1 Report

A very interesting article addressing two points of view of authors who have dedicated their research to machines, computers and AI while making a direct link with multidisciplinary artistic practices. Relevant text for any researcher interested in these issues.

Author Response

We thank the reviewer for their appreciation of this essay,

Reviewer 2 Report

This is an exciting topic to discuss. The examples illustrated in the article are intriguing. I found the videos of the projects listed in the article, including Paul the Robot, to be fascinating. I recommend publication with minor revisions.

Recommendations:

The statements of author 2 between lines 76 and 81 are conflicting and irrelevant to the comments of author 1 above and below. Author 2 says :” In this regard, its useful to compare your views to Howard Gardner’s (Gardner 1983) theory of multiple intelligences (MI).5 One of his criteria for ‘an intelligence’ is that it can be found in non-human species. Furthermore, he considers intelligence as: the capacity to solve problems or make products of value in one or more cultural settings (Gardner 1983, p.x). This definition differs from intelligence as defined by standard IQ tests, which focus exclusively on linguistic and logical/mathematical abilities. Gardner’s more inclusive definition of intelligence emphasizes the arts as domains of intelligence, just as yours does.” However, Gardner and author 1's ideas are conflicting; hence, author 2's statement "just as yours does" is inaccurate.

Author 2 says in lines 91-95. “I have a similar opinion about drawing, that it is one of the few uniquely human capacities. Drawing is the original graphic symbol system and the root of later systems including writing, numbers, and musical notation. Thus I find it ironic that drawing is now largely neglected as part of childhood education.”

First, drawing is not an exclusively human ability. There are trained animals who can sketch. Secondly, drawing is not largely neglected as part of childhood education. Many artists practice drawing.

Some minor edits:

Line 10, 68: sapiens appeared twice

Author Response

We thank the reviewer for their appreciation of the work and their recommendations for improving the text. We have addressed the points raised the best we can.